environmental science/ecology/acoustics

acoustics, wildlife monitoring, conservation, cetaceans, echolocation, *Tursiops truncatus*

**Author for correspondence:**
Bianca Romeu
e-mail: romeu.bianca@gmail.com

# Low-frequency sampling rates are effective to record bottlenose dolphins

Bianca Romeu[1,2], Alexandre M. S. Machado[1,2,4], Fábio G. Daura-Jorge[1,2], Marta J. Cremer[2,3,5], Ana Kássia de Moraes Alves[5] and Paulo C. Simões-Lopes[1,2]

[1]Laboratório de Mamíferos Aquáticos, Departamento de Ecologia e Zoologia, and [2]Programa de Pós-Graduação em Ecologia, Universidade Federal de Santa Catarina, Florianópolis, Brazil
[3]Programa de Pós-Graduação em Saúde e Meio Ambiente, Universidade da Região de Joinville, Joinville, Brazil
[4]Department of Collective Behaviour, Max Planck Institute of Animal Behaviour, Konstanz, Germany
[5]Laboratório de Ecologia e Conservação de Tetrápodes Marinhos e Costeiros, Universidade da Região de Joinville, Joinville, Brazil

BR, 0000-0001-7911-1237; AMSM, 0000-0001-6252-6890; FGD-J, 0000-0003-2923-1446; MJC, 0000-0003-3521-1409; AKM, 0000-0003-1195-4603; PCS-L, 0000-0002-7338-3669

Acoustic monitoring in cetacean studies is an effective but expensive approach. This is partly because of the high sampling rate required by acoustic devices when recording high-frequency echolocation clicks. However, the proportion of echolocation clicks recorded at different frequencies is unknown for many species, including bottlenose dolphins. Here, we investigated the echolocation clicks of two subspecies of bottlenose dolphins in the western South Atlantic Ocean. The possibility of recording echolocation clicks at 24 and 48 kHz was assessed by two approaches. First, we considered the clicks in the frequency range up to 96 kHz. We found a loss of 0.95–13.90% of echolocation clicks in the frequency range below 24 kHz, and 0.01–0.42% below 48 kHz, to each subspecies. Then, we evaluated these recordings downsampled at 48 and 96 kHz and confirmed that echolocation clicks are recorded at these lower frequencies, with some loss. Therefore, despite reaching high frequencies, the clicks can also be recorded at lower frequencies because echolocation clicks from bottlenose dolphins are broadband. We concluded that ecological studies based on the presence–absence data are still effective for bottlenose dolphins when acoustic devices with a limited sampling rate are used.

# 1. Introduction

Acoustic behaviour is a crucial element of many marine species. This is particularly the case for cetaceans which use acoustics for practically all activities. Odontocetes use whistles and clicks. Whistles— narrowband and frequency-modulated signals—are used to transmit information, for individual recognition and group cohesion [1–3]. Clicks—pulsed sounds—are used during socialization [2,4], navigation and foraging [5,6]. Cetaceans spend a small proportion of their time at the surface and travel kilometres in a couple of hours. Therefore, the use of acoustic methods to investigate and record acoustic emissions is an effective way to study their ecology [7].

Passive acoustic monitoring (PAM) has been successfully used in ecological studies [8–10]. It enables extensive geographical areas to be monitored, from hours to months [11], and reveals the occupancy patterns in space and time [12]. Such studies can improve conservation and management plans, determining the presence of dolphins in marine protected areas or areas important for fishery activities, investigate their habitat use, monitor their responses to anthropogenic activities, and estimate population parameters [8,13–17]. When using PAM, however, researchers need to make decisions about the most appropriate acoustic devices and their sampling protocol. These are based on the research aims and budget, and target species particularities, such as the amplitude of their acoustic emission frequencies [18,19].

For odontocetes, echolocation clicks are the key emissions recorded by PAM because they are used in their most frequent behaviours, such as navigation and foraging [20–23]. Echolocation clicks can be produced over a wide frequency range, and acoustic devices with high sampling rates, ranging up to 576 kHz, may be important in some situations [8,14,24]. These acoustic devices can be expensive [7], which constrains the use of PAM when the study budget is limited [25,26]. However, when the aim of the study is only to detect the presence–absence patterns (i.e. distribution, occupancy and use of habitat), an acoustic device that, even partially, records clicks in the lower frequencies, can be a cost-effective alternative. To use these lower-frequency devices, it is crucial to know the possibility of recording clicks at different frequencies.

The minimum sampling rate required for echolocation acoustic records also depends on the target species. Some cetacean species, as *Kogia* spp. and *Pontoporia blainvillei*, produce narrowband high frequency (NBHF) echolocation clicks [27,28], which often restrict the minimum sampling rate to higher frequencies. By contrast, if clicks have a wide frequency range, they can be recorded at lower frequencies. For instance, for some species of the Delphinidae family, such as the common dolphins (*Delphinus delphis*), the detection rate of burst pulse clicks was similar between sampling rates at 96, 192 and 300 kHz, and the burst pulse acoustic parameters were also similar between those frequencies [29]. This indicates that the use of more cost-effective devices, that do not record higher frequencies, may be effective for at least some species. Furthermore, the use of lower sampling rates would improve the storage and processing of the large volume of data collected during a PAM [29].

The use of lower frequencies also seems a possibility for research on *Tursiops truncatus*, which has broadband echolocation clicks [30–32]. Their echolocations occur from audible frequencies (less than 20 kHz) to at least 150 kHz [33]. However, there are no studies that describe the proportion of echolocation clicks that occur in different frequencies through this frequency range, or how efficient lower sampling rates are to record these clicks of free-living animals. Here, we evaluated whether the echolocation clicks produced by the bottlenose dolphins can be effectively recorded in frequencies up to 24 and 48 kHz. We aim to promote the use of PAM in ecological research on bottlenose dolphins, because it is a useful tool that can inform conservation and management plans, especially in countries where science budgets are limited.

# 2. Methods

## 2.1. Data sampling

Echolocation clicks of two bottlenose dolphin subspecies that occur across different environments in the western South Atlantic Ocean (wSAO) were analysed. These were *Tursiops truncatus truncatus*, which are found in open waters, and *T. t. gephyreus*, which are found in coastal areas. We collected *T. t. gephyreus* data from the lagoon system adjacent to Laguna (28°20′ S, 48°50′ W), southern Brazil (figure 1). This lagoon system has depths of between 0.4 and 13 m, with an average depth of 1.8 m. The data were collected from 4 to 12 December 2017, using a 4.4 m inflatable research boat with a 30 hp outboard engine. The recordings were made with the engine off at sites with depths of 2.5–5.0 m. The hydrophone was positioned 1.5 m from the surface and 15 groups were sampled using recordings that

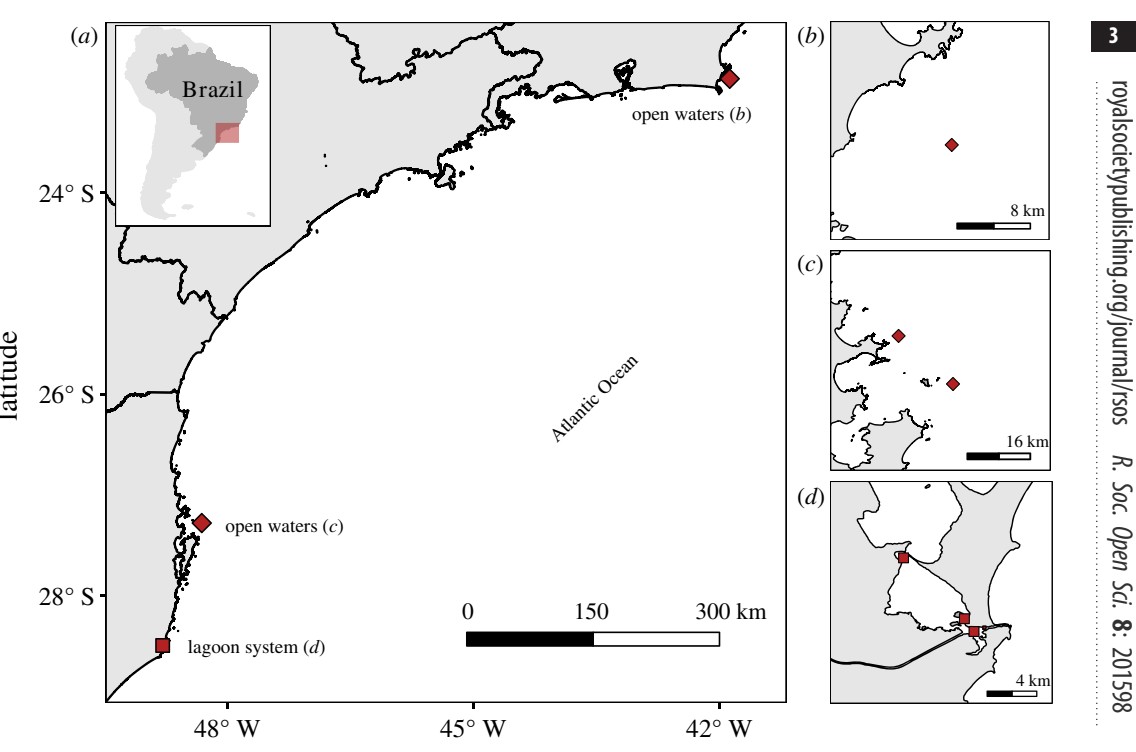

**Figure 1.** Data collection sites in the Brazilian coast of the western South Atlantic Ocean (*a*). Recording sites of *Tursiops truncatus truncatus* in open waters (*b,c*) and recording sites of *Tursiops truncatus gephyreus* in a lagoon system in southern Brazil (*d*). Coordinates were re-projected to WGS84 datum.

lasted approximately 4–34 min. Each sampled group contained two to seven dolphins that were displaying travel or foraging behaviour (electronic supplementary material, table S1).

The *T. t. truncatus* were opportunistically recorded in open waters by the Cetaceans Project Monitoring in the Santos Basin area, south and southeastern Brazil. The data were collected on 14 and 24 February 2017, from three sites near the coast, at depths of 21, 28 and 35 m (figure 1 and electronic supplementary material, table S1). Photographic records confirmed the subspecies *T. t. truncatus* [34]. The group sizes ranged from 12 to 30 individuals, and two groups showed travelling behaviour, and one was travelling/foraging. The recordings lasted approximately 7–16 min (electronic supplementary material, table S1). Two types of vessels were used: a 23.7 m mini supply vessel with two 325 hp engines and a 5 m inflatable boat with a 50 hp outboard engine. The hydrophone was positioned 5 m from the surface and the engines were off during the recordings.

All the recordings from both subspecies were made using a Reson TC 4032 hydrophone (0.005–120 kHz) connected to a Sony PCM-D100 recorder with a sampling rate of 192 kHz/24 bit (maximum frequency of 96 kHz – Nyquist frequency).

## 2.2. Sample processing

Two different approaches were used to analyse if echolocation clicks recorded in a frequency range up to 96 kHz can be recorded in frequency ranges below 24 and 48 kHz. First, the recordings were assessed visually through spectrograms to quantify the proportion of echolocation clicks occurring in each frequency range of interest. That is, each click observed in the spectrograms was verified if it occurred below and above frequency thresholds of 24 and 48 kHz. Second, the recordings were downsampled at 48 and 96 kHz (Nyquist frequency of 24 kHz and 48 kHz, respectively) and processed by an automatic signal detector, to test if echolocation clicks are recorded when recordings are made at a lower sampling rate.

### 2.2.1. Proportion of echolocation clicks in each frequency range

The recordings from each dolphin group were fractionated into 1 min samples to standardize the sample units. However, some recording durations were longer than multiples of 1 min. Therefore, to use all

records, we included samples of less than 1 min in the analysis, that is, those sections beyond the 1 min limit of the last sample. We analysed the samples in spectrograms using Raven Pro 1.6.1 software, with a sampling rate of 192 kHz/24 bit, Hann window, 512 points in size and overlap of 50%. Raven uses Fourier transform to create spectrograms as a frequency domain representation of the signal.

The spectrograms were visually inspected to identify echolocation clicks. These clicks were defined and identified as those belonging to click trains with inter-click intervals longer than 10 ms [9,35,36]. The total number of echolocation clicks recorded up to 96 kHz was counted manually. Each echolocation click was visually inspected to verify its occurrence below and/or above frequency thresholds of 24 and 48 kHz given the full frequency range of the recordings. Then, clicks were counted in each frequency threshold to estimate the proportion of clicks that appear in each threshold. The total number of echolocation clicks recorded up to 96 kHz was paired with the total number of echolocation clicks that occurred in 24 and/or 48 kHz thresholds (figure 2). This is because the same click, when visualized below 24 kHz was counted as 'up to 24 kHz' and 'up to 48 kHz' (figure 2, the solid line brown rectangles), while the clicks with frequencies above 24 kHz, but below 48 kHz, were only counted as 'up to 48 kHz' (figure 2a, the dashed line brown rectangle).

### 2.2.2. Downsampling and automatic detections

A sampled high-frequency signal can change when it is sampled in different, lower sampling rates [30]. Because of this, in our second approach, the recordings were downsampled at 48 and 96 kHz to analyse if echolocation clicks are recorded at 24 and 48 kHz, using an automatic signal detector. The downsample and the following automatic clicks detection were made using R 3.6.0 [37]. First, a fourth-order Butterworth 15 kHz high-pass filter and an anti-aliasing finite impulse response (FIR) low-pass filter to 24 or 48 kHz were applied in the original recordings, for each corresponding downsample frequency. Then, the downsample was made using the 'downsample' function of the 'tuneR' R package [38]. Next, we used the 'auto_detec' function of the 'warbleR' R package [39] to detect the echolocation clicks automatically in recordings at 96, 48 and 24 kHz—in other words, the original and downsampled frequencies. The parameters to configure the 'auto_detec' were defined through the 'optimize_auto_detec' function from the 'warbleR' R package [39].

The 'optimize_auto_detec' function takes a selection table containing the times of each signal (echolocation clicks, in this case), and then runs the automatic detection with multiple parameters to find the ones that maximize sensitivity and specificity of signal detections compared to the selection table. Subsamples of 5 s from different recordings and different dolphins' groups were used to select the echolocation clicks and validate automatic detections. The selection tables contained one subsample of each *T. t. truncatus* group (a total of three groups), and one subsample of two different *T. t. gephyreus* groups. The signals were manually selected in the subsamples of each sampling rate, to define the best-adjusted parameters to detect clicks in each frequency (96, 48 and 24 kHz). The least number of subsamples from *T. t. gephyreus* used was due to these dolphins being from the same population and environment (lagoon system).

Even though we have selected the optimal parameters to configure the signal detector based on *a priori* manual detections, the sensibility and specificity of the detector were not the same for all recordings. Then, only the presence/absence of signal detections were considered, since we cannot guarantee that all detections were true positives. Furthermore, additional filters were applied to reduce the number of false-positive detections. First, all detections were filtered based on the interval of signals detected, similar to inter-click intervals (see [40]), excluding detections with an interval longer than 0.2 s, since echolocation clicks occur in click trains, not isolated. Second, the detection localization (in time) in the original recording was compared with detection localization in downsampled recordings, assuming that the detector performance was better in recordings at 96 kHz. Since the echolocation clicks in the downsampled recordings tend to be longer than in the original record, a buffer was created around each detection of the original recordings by expanding the start and end time of detections by the length of each signal (i.e. click duration) to guarantee the match between the same detections at each frequency. Next, recordings were divided into bins of one second and matching bins from the division of the 1 min samples from the first approach.

The bins from the original recordings (96 kHz) were binarized. Only 1 s bins with at least one signal detected present were kept and compared with the same respective bins in 24 and 48 kHz. Finally, the matches between bins with the presence of signals in downsampled recordings (24 and 48 kHz) and in original recordings (96 kHz) were quantified. Finally, this yields a proportion of seconds with signals detected in downsampled recordings given the total number of seconds with detections in the original files. These proportions in each 1 min sample (or fractions) were used to calculate the probability to detect echolocation clicks at 24 or 48 kHz.

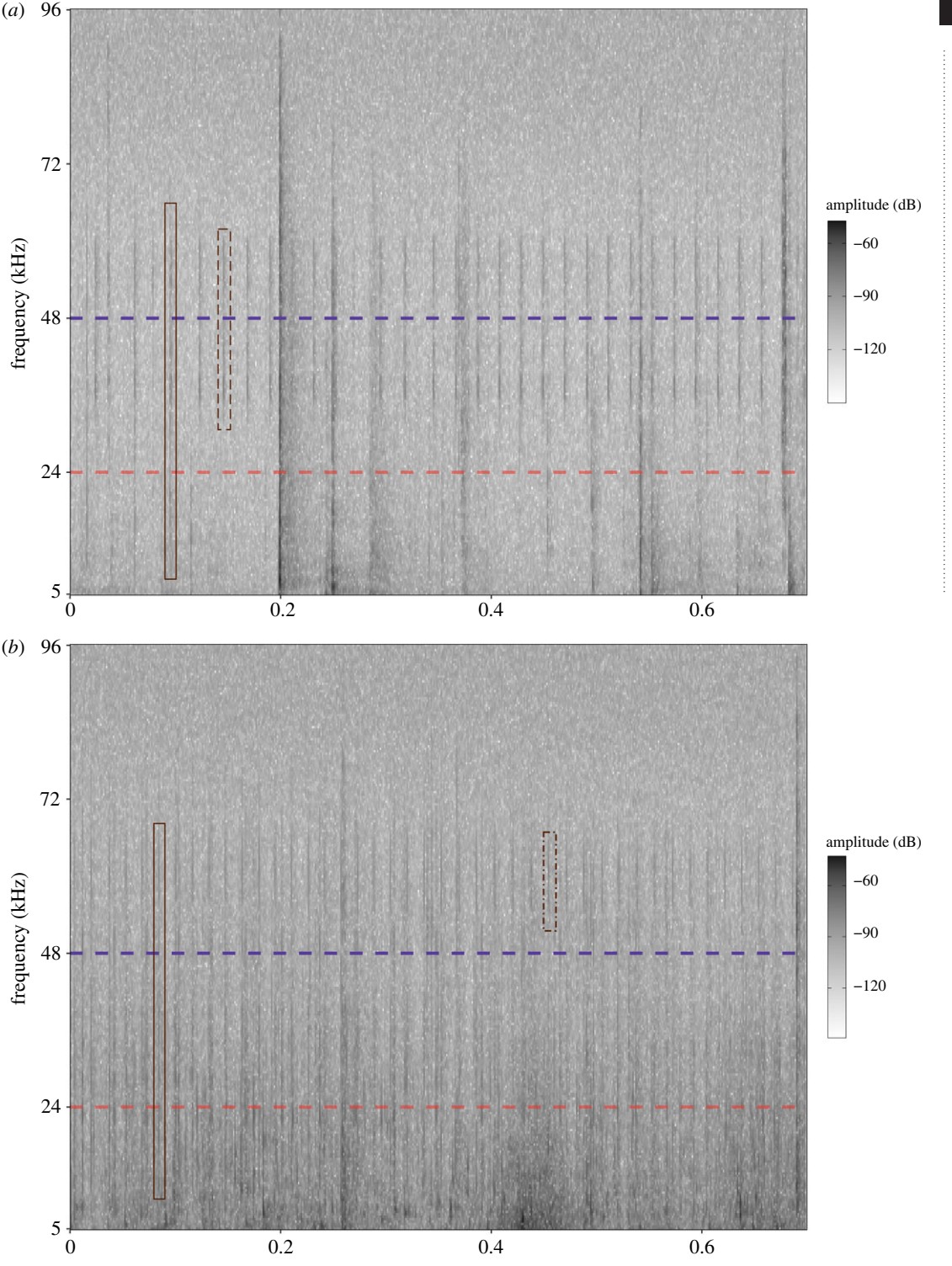

**Figure 2.** Spectrograms of a sample subset show the broadband frequency (kHz) of bottlenose dolphins echolocation clicks over time. Dashed lines are colour coded to define the 24 and 48 kHz frequency thresholds. Brown rectangles indicate echolocation clicks detected below 24 and 48 kHz (solid line), clicks detected below 48 kHz and above 24 kHz (dashed line in *a*), and clicks detected above 48 kHz (dot-dash line in *b*).

## 2.3. Sampling rate assessment

First, the hypothesis that bottlenose dolphins' echolocation clicks occur equally in frequency ranges of up to 24, 48 and 96 kHz, when recorded from free-living animals, was tested. A generalized linear mixed-

effects model (GLMM) was constructed using a binomial error structure with a logit link function. The proportion of echolocation clicks that occur at each frequency range out of the total number of echolocation clicks recorded up to 96 kHz was the response variable and the frequency range and environment (lagoon system or open water) were fixed effect terms. To account for sample pairing (up to 24 and 48 kHz), pairs of samples were treated as random effects and were used nesting within recording identities to account for autocorrelation in group composition and behavioural context during sampling.

A candidate model was built to test the hypothesis and it was compared to a null model containing only the intercept. Models were compared using the Akaike information criterion corrected (AICc) for small samples and the Akaike weights [41]. The model with the lowest AICc value and the highest Akaike weight was considered to be the most parsimonious and the one that better supported data variation [41]. We simulated 10 000 datasets from the fitted model to validate the model assumption. Then, the Kolmogorov–Smirnov test was used to determine whether the deviations between the observed and expected residual distributions were significant (electronic supplementary material, figure S1; for details about validation methods, see [42]). The random effects were validated by visually comparing a QQ-plot of the random effects quantiles against the standard normal quantiles (electronic supplementary material, figures S2 and S4). The assumptions were valid if most values fell along the line. Model fit was assessed using the theoretical marginal and conditional $R^2_{GLMM}$ [43]. Marginal $R^2_{GLMM}$ represents the proportion of the total variance explained by the predictors (fixed effects) and conditional $R^2_{GLMM}$ represents the proportion of the variance explained by both the fixed and random effects.

Finally, the hypothesis that the echolocation clicks are recorded at frequencies up to 24 and 48 kHz, was tested. A second set of GLMM was constructed using a binomial error structure to test if echolocation clicks can be detected equally across the different sampling rates (the downsampled recordings). Considering only the bins with the presence of at least one signal in the original frequency, the response variable was the proportion of bins with detections that occur in the 1 min samples at each downsampled frequency out of the total number of bins with detections in the equivalent sample in original frequency. The frequency and the environment (lagoon system or open water) were used as predictors, similar to the first model. The 1 min samples were treated as random effects, nested within recording identities to account for autocorrelation in group composition and behavioural context during sampling. The next steps in this analysis, which involved candidate model build, models comparison, and model fit assessment, were conducted as in the first model. All analyses were conducted using R 3.6.0 [37]. R Code used in the analyses is available in https://datadryad.org/stash/share/s9_1N6tdK9pJatiufR-2Qv75Uo7f9Ym3sGp5hrG-HH4.

# 3. Results

A total of 3 h 16 min were recorded, 2 h 45 min from *T. t. gephyreus* and 31 min from *T. t. truncatus*, which resulted in 196 samples (158 lasting 1 min and 38 lasting less than 1 min). A total of 74 187 echolocation clicks were counted manually in the 96 kHz frequency range. Echolocation clicks occurred below the 24 and 48 kHz frequency ranges in all samples analysed visually (table 1). A total of 8527 1 s bins with detection present from original recordings were compared with the same bins from 24 and 48 kHz downsampled recordings.

## 3.1. Probabilities of recording echolocation clicks below 24 and 48 kHz

To our first approach, using spectrograms, the most parsimonious model explaining differences in the likelihood of occurrence at the full range of echolocation clicks included an interaction between the environment and frequency range (table 2). The 24, 48 and 96 kHz frequencies were not equally efficient. The likelihood of occurrence of echolocation clicks at the frequency range up to 24 kHz was lower than that up to 48 kHz in both the lagoon system (log odds ratio = −5.2, s.e. = 0.104, d.f. = 386, $t = -50.430$, $p < 0.001$) and open waters (log odds ratio = −3.655, s.e. = 0.084, d.f. = 386, $t = -43.538$, $p < 0.001$). Our model predicted a mean error in the occurrence of only 0.95% in the frequency range up to 24 kHz (mean probability = 99.05% ± 0.003 s.e.; CI = 98.3–99.46%) and 0.01% in 48 kHz (mean probability = 99.99% ± 0.00001 s.e.; CI = 99.99–99.99%) in the lagoon system (figure 3). In open waters, we observed a mean error of 13.9% at 24 kHz (mean probability = 86.1% ± 0.061 s.e.; CI = 69.26–94.45%) and only 0.42% at 48 kHz (mean probability = 99.58% ± 0.002 s.e.; CI = 98.85–99.85%) (figure 3;

**Table 1.** Sample size per group and descriptive results of clicks counted manually. Samples where echolocation clicks that occurred below 24 and 48 kHz frequencies (Samples with occurrence), and the total echolocation clicks occurrence below each of these frequencies (Total clicks observed) on each analysed subspecies in their environment, lagoon system and open waters. The total variation of clicks counted in each frequency (min–max clicks counted).

| subspecies/environment | frequency threshold (kHz) | samples with occurrence (N observed/total) | total clicks observed (Mean ± s.d.) | min–max clicks counted |
|---|---|---|---|---|
| T. t. gephyreus/lagoon system | 24 | 159/159 | 44 546 (280 ± 293) | 7–1668 |
| T. t. gephyreus/lagoon system | 48 | 159/159 | 49 627 (312 ± 336) | 7–2163 |
| T. t. truncatus/open waters | 24 | 37/37 | 19 932 (539 ± 528) | 10–2120 |
| T. t. truncatus/open waters | 48 | 37/37 | 24 028 (649 ± 597) | 13–2243 |

**Table 2.** Comparison between null models and the candidate models for the proportion of clicks recorded in different frequency ranges and reduced sampling rates. Predictors included in binomial generalized linear models are shown in the columns 'Env' (environment), 'Freq' (frequency threshold) and 'Env:Freq', which represents the interaction term between the two predictors. The response variables were the proportion of echolocation clicks recorded in each frequency range out of the total clicks recorded/ detected up to 96 kHz (proportion of echolocation clicks in each frequency range), and the proportion of seconds with signals detected in downsampled recordings given the total number of seconds with detections in the original files (proportion of detection in each frequency). Models are ranked by AICc (Akaike information criteria corrected for small samples) and presented along with the degrees of freedom (d.f.), log-likelihood (log-like), the change in AICc relative to the best model (ΔAIC) and Akaike weights.

| model | intercept | Env | Freq | Env:Freq | d.f. | log-like | AICc | ΔAICc | Akaike weight |
|---|---|---|---|---|---|---|---|---|---|
| proportion of echolocation clicks in each frequency range | | | | | | | | | |
| M1 | 4.650 | + | + | + | 6 | −1404 | 2820 | 0 | 1 |
| null model | 4822 | − | − | − | 3 | −8435 | 16 876 | 14 056 | 0 |
| proportion of detection in each frequency | | | | | | | | | |
| M1 | 0.8532 | + | + | + | 6 | −1291.515 | 2595.2 | 0 | 1 |
| null model | 1.6240 | − | − | − | 3 | −2454.155 | 4914.4 | 2319.17 | 0 |

see electronic supplementary material, table S2 for more details on the comparisons). Such fixed effects accounted for 56.70% (marginal $R^2_{GLMM}$) of the differences in the number of echolocation clicks detected, and 99.94% (conditional $R^2_{GLMM}$) when both the fixed and random effects were considered.

## 3.2. Probability of detecting clicks in downsampled recordings

To the approach using the automatic signal detector, the candidate model had better support than the null model (table 2). The 24, 48 and 96 kHz frequencies were not equally efficient. The likelihood of detecting the signals from the original frequency (96 kHz) at 24 kHz was lower than that at 48 kHz in both the lagoon system (log odds ratio = −2.089, s.e. = 0.054, d.f. = 558, $t$ = −38.241, $p < 0.001$) and open waters (log odds ratio = −1.878, s.e. = 0.137, d.f. = 588, $t$ = −13.620, $p < 0.001$). Our model predicted a mean error in detection of 29.88% at 24 kHz (mean probability = 70.12% ± 0.02 s.e.; CI = 66.14–74.1%) and 5.01% in 48 kHz (mean probability = 94.99% ± 0.005 s.e.; CI = 93.99–95.99%) in the lagoon system (figure 4). In open waters, we observed a mean error of 12.37% at 24 kHz (mean probability = 87.62% ± 0.023 s.e.; CI = 82.95–92.3%) and 2.11% at 48 kHz (mean probability = 97.88% ± 0.005 s.e.; CI = 96.88–98.89%) (figure 4; see electronic supplementary material, table S3 for more details on the comparisons). Such fixed effects accounted for 55.7% (marginal $R^2_{GLMM}$) of the differences in the number of detections, and 95.9% (conditional $R^2_{GLMM}$) when both the fixed and random effects were considered. The scaled residuals slightly deviate from the expected distribution (electronic supplementary material, figure S3).

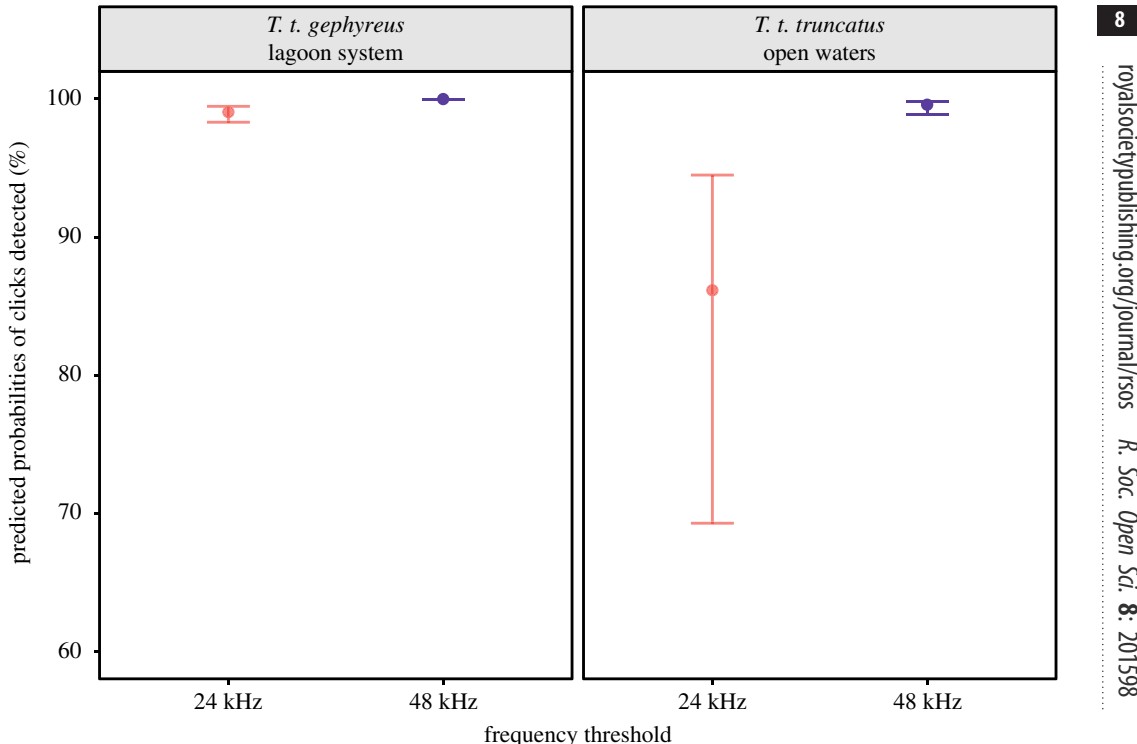

**Figure 3.** Predicted probabilities of echolocation clicks recorded below 24 and 48 kHz. Dots and whiskers indicate the estimated marginal means for each frequency range and the 95% confidence intervals, colour coded by the frequency range. Estimated marginal means are back-transformed from the logit scale.

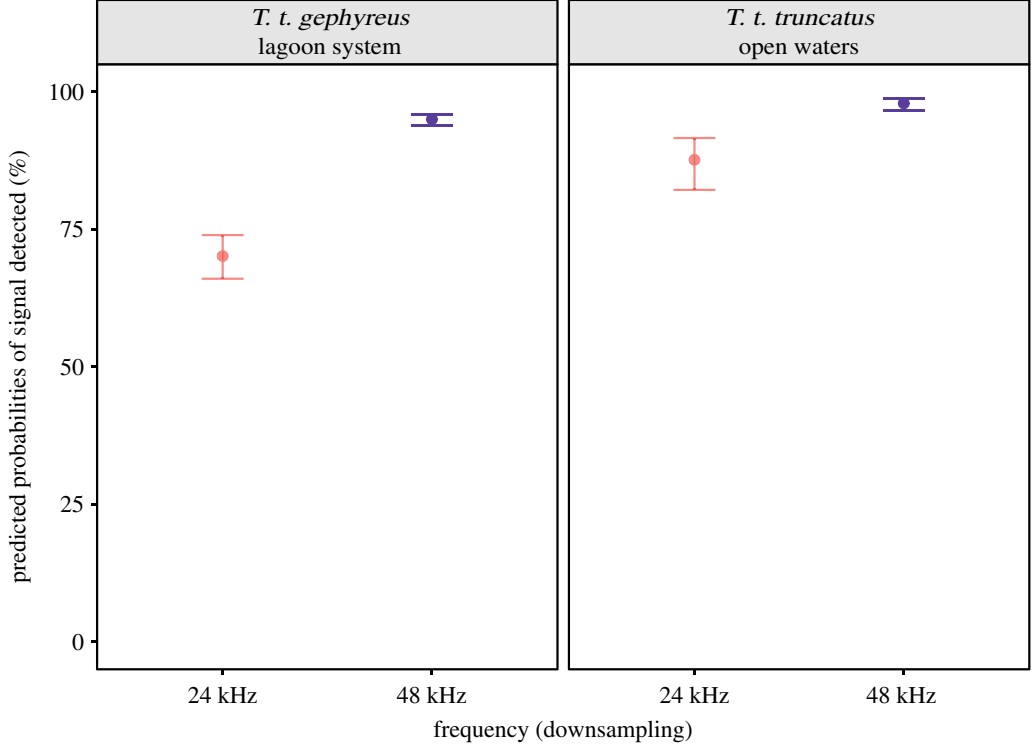

**Figure 4.** Predicted probabilities of detecting signals in downsampled recordings below 24 and 48 kHz. Dots and whiskers indicate the estimated marginal means for each frequency and the 95% confidence intervals, colour coded by the frequency. Estimated marginal means are back-transformed from the logit scale.

# 4. Discussion

We show for the first time that most of the echolocation clicks of free-living *Tursiops truncatus* recorded up to 96 kHz also occur in frequency ranges below 24 and 48 kHz. In downsampled recordings, echolocation clicks are also detected below these frequencies. This showed that *T. truncatus* often produce echolocation clicks in a frequency range that can still be recorded with reduced sampling rates. These findings showed that acoustic devices that only record maximum frequencies of 24 and 48 kHz could be used to investigate ecological aspects of bottlenose dolphins, such as distribution and habitat use [44,45]. These reduced frequencies could improve PAM programmes, reduce equipment costs and facilitate data storage and management. Ultimately, this research will benefit bottlenose dolphin conservation in developing countries or where PAM programmes to study this species are compromised by costly equipment.

Our estimate of the proportion of echolocation clicks recorded with frequency ranges below 24 and 48 kHz shows that these clicks can be recorded at lower frequencies. However, this proportion cannot be extrapolated to the echolocation clicks recorded at lower sampling rates. When a high-frequency signal is sampled with a low sampling rate, the sampled signal can be different from the real signal [30]. Then, we downsampled the recordings to simulate recordings made with lower sampling rates and confirm that echolocation clicks are still preserved in these frequencies. Therefore, unfortunately, we cannot present a definitive evaluation of the performance of lower sampling rates in recording echolocation clicks because of the effect of three main reasons discussed below: frequencies evaluated, soundscape and detector performance.

Downsampling the original files at 48 and 96 kHz sampling rates increase the soundscape interference in the signal detection. Snapping shrimp snaps, for example, are commonly present in the soundscape, and their sound is similar to dolphin clicks [46–49]. In our study area, mainly in the lagoon system, snapping shrimp snaps are part of the soundscape. Then, when we used the automatic detector in the downsampled recordings, especially at 24 kHz, the detection performance was limited. It is not difficult to differentiate the echolocation clicks from the snapping shrimp snaps when visually inspecting spectrograms. The former has patterns in inter-click intervals [46]. However, the soundscape composition can limit the use of an automatic detector [50]. To avoid counting shrimp snaps, we could exclude as much as possible the false-positive detections using some additional steps in our processing protocol. However, there is not a definitive solution to this problem [50], and the use of automatic detector at lower sampling rates has some limitations.

Despite these limitations considered, our main goal here was to test if lower frequencies can be used to record the presence of bottlenose dolphins through echolocation signals. Analysing the downsampled recordings, we showed that echolocation clicks are recorded even in recordings at low sampling rates. Therefore, our results, from both frequency range and detection analysis, show that echolocation clicks emissions from *T. truncatus* can be recorded using at least 24 kHz (sample rate of 48 kHz) and a 1 min sample size is sufficient to record the clicks. However, there were differences in the proportion of echolocation clicks that occur and the probability of detection below 24 kHz between the two subspecies.

Analysing the frequency ranges, we found that the chance of echolocation clicks from *T. truncatus truncatus* occurring below 24 kHz was smaller and more inaccurate, although clicks were visualized below this frequency in all samples. We decided to analyse two subspecies in two different environments, even with few records of *T. t. truncatus*, because the acoustic parameters of echolocation clicks can vary between different *Tursiops* species [51] and environmental characteristics can affect sound production and propagation [5,52,53]. However, we were unable to distinguish whether the reduction in the occurrence of echolocation clicks from *T. t. truncatus* below 24 kHz is due to environmental variations, differences in sound emissions between subspecies, or both, because each subspecies inhabits a different environment and can adapt their sound emissions according to environmental characteristics [54]. Furthermore, the random effects of our models show that other factors than the subspecies and the environment can also explain such variations in the frequency range of echolocation clicks. Many of these factors cannot be controlled when we record free-living animals, such as the distance and position of a moving dolphin to the hydrophone [55]. However, more samples from *T. t. truncatus* can help us to understand this pattern, as well as consider other variables such as behaviour, group size and composition, and environmental conditions.

The results from the automatic detector showed that clicks are more likely to be detected in recordings at 24 kHz from *T. t. truncatus* than in recordings from *T. t. gephyreus* at the same frequency. This result can be representing the effect of the soundscape. Since the soundscape in our samples from the open waters had fewer snapping shrimp snaps, the detection performance in this environment was quite better. Despite the limitation aforementioned, the general results from the automatic detections reinforces our findings using manual inspection of the frequency ranges: recordings at 48 kHz are better than 24 kHz to record or detect echolocation clicks, but 24 kHz can

still record echolocation clicks. In this respect, our results also indicated frequencies at which the acoustic repertoire of *T. truncatus* can be recorded.

By showing that echolocation clicks could still be recorded and detected below 48 kHz, our results reinforce the idea that most sounds made by *T. truncatus* can be recorded at low frequencies, even below 24 kHz [56]. Although some burst pulsed sounds and echolocation clicks are exclusively ultrasonic [56], this seems to be uncommon. Burst pulsed sounds that are formed by broadband pulses in a similar way to echolocation signals, generally present frequency components in the audible frequency range [57–59]. The components of echolocation clicks that have low frequencies appear when recordings are made outside of the transmitting beam of the dolphin. Echolocation clicks are emitted in a directional beam forward from the melon in the dolphin head. High-frequency echolocation clicks are recorded mainly when recordings are made on the axis of the beam [5]. Outside the beam, the low-frequency portion of echolocation clicks suffer less attenuation and lower-frequency components are introduced in the click frequency spectrum [60–62]. Therefore, it is expected that low-frequency elements become more common when recording free-living animals because these recordings are mainly off-axis of the dolphin transmitting beam. Finally, *T. truncatus* whistles usually occur at frequencies up to 30 kHz [59,63]. Specifically, in the *T. t. gephyreus* population sampled, less than 1% of whistles had part of the fundamental frequency above 24 kHz [64]. Therefore, we can conclude that all sounds produced by *T. truncatus* can be recorded using a frequency of 48 kHz. It is also possible to use a frequency of 24 kHz if some information loss is acceptable.

Here, we showed that lower frequencies record bottlenose dolphins' echolocation clicks efficiently for a set of ecological studies. In other words, studies that investigate the occurrence or distribution of individuals in a population, when the presence/absence data are sufficient, can use recordings at low frequencies. However, it is necessary to highlight, when the goal is to analyse acoustic parameters of clicks or identify different dolphins' species, it is recommended to record, as much as possible, all clicks frequency ranges [51,65]. Also, one should consider the soundscape of the study area, thinking about possible solutions to the acoustic data analysis.

Despite the limitations discussed here, the effectiveness of reduced sampling rates when recording echolocation clicks should stimulate the acoustic monitoring of bottlenose dolphins, and maybe other cetaceans, in developing countries where the available methods for studying marine mammals remain limited due to budget restrictions [26]. Despite being an effective way to study cetaceans [19,66], PAM is still expensive [25]. Inexpensive alternatives with modular hardware and open-source software exist for PAM in terrestrial environments [67], but adaptations to record high frequencies are still required. Recently, a low-cost acoustic device has been developed with sampling rates up to 384 kHz [68]. Its new version permits connecting a hydrophone to it. It can be a solution to implement a PAM project with a limited budget, but the costs can be higher when considering import taxes and shipping costs. Although it is necessary to investigate other dolphins species, showing that *T. truncatus* echolocation clicks are effectively recorded at lower frequencies will encourage the use of alternative acoustic devices with lower sampling rates. This will help to alleviate the prohibitive cost faced by researchers in developing countries even though some compromises need to be accepted when using lower sampling rates [19].

# 5. Conclusion

We conclude that a frequency of 48 kHz is effective for recording the entire acoustic repertoire of *T. truncatus*, including echolocation clicks. However, it is also possible to use 24 kHz to record *T. truncatus* if some information loss is acceptable. Furthermore, 1 min sample size is sufficient to record echolocation clicks with frequencies of 24 and 48 kHz, considering the limitations of these frequencies. Ultimately, our results can expand the use of PAM as an accessible tool to ecological studies that can help conserve bottlenose dolphins, especially *T. t. gephyreus*, an endemic subspecies that occurs from southern Brazil to central Argentina and has recently been categorized as vulnerable [69].

Ethics. Research approved by the Brazilian Ministry of the Environment, permit ABIO no. 657/2015 (Cetacean Monitoring Program) and SISBio no. 64986 (*Tursiops truncatus gephyreus* data collected in a Lagoon System).
Data accessibility. Data and R code to reproduce the results are available at Dryad Digital Repository: https://doi.org/10.5061/dryad.4f4qrfj99 [70].
Authors' contributions. Conceptualization: B.R., F.G.D.-J. and P.C.S.-L.; Data curation: B.R.; Formal analysis: B.R. and A.M.S.M.; Funding acquisition: B.R. and F.G.D.-J.; Data collection: A.K.M.A.; Methodology: B.R., A.M.S.M., F.G.D.-J. and A.K.M.A.; Project administration: B.R., F.G.D.-J. and P.C.S.-L.; Writing—original draft: B.R. and A.M.S.M.; Writing—review and editing: B.R., A.M.S.M., F.G.D.-J., M.J.C. and P.C.S.-L. All authors read and approved the final manuscript.

Competing interests. We declare we have no competing interests.

Funding. This study was financed in part by the Coordenação de Aperfeiçoamento de Pessoal de Nível Superior – Brasil (CAPES) – Finance Code 001 (B.R.; A.M.S.M.).

Acknowledgements. F.G.D.-J., P.C.S.L. and M.J.C. thank CNPq for the research productivity scholarships (308867/2019-0, 305573/2013-6 and 10477/2017-4, respectively). F.G.D.-J. was also supported by CAPES (#88887.374128/2019-00). We are very grateful to Natanael Silva and Pedro V. Castilho for data collected in Laguna, and Renan L. Paitach for data collected in open waters. We thank Robin C. Whytock and James Christie for their help with the Solo audio recorder. We thank Professors Emygdio L.A. Monteiro-Filho, Renato H.A. Freitas, Eduardo L.H. Giehl, Guilherme R.R. Brito and the two anonymous reviewers for important considerations about the text. We thank Socioambiental Consultores Associados for kindly lending the recorder used for data collecting. We thank Petrobrás for kindly authorizing the use of data collected during the Projeto de Monitoramento de Cetáceos na Bacia de Santos (PMC-BS). The PMC-BS is one of the monitoring programmes required by Brazil's federal environmental licensing process of the oil production and transport by Petrobrás at the Santos Basin pre-salt province (process no. 02001.114279/2017-80, ACCTMB no. 657/2015).

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
