## [Peer Review File · Royal Society Open Science]

Review History

RSOS-201598.R0 (Original submission)

Review form: Reviewer 1

Is the manuscript scientifically sound in its present form?

Yes

Are the interpretations and conclusions justified by the results?

No

Is the language acceptable?

Yes

Do you have any ethical concerns with this paper?

No

Have you any concerns about statistical analyses in this paper?

No

Recommendation?

Major revision is needed (please make suggestions in comments)

Comments to the Author(s)

I have two major concerns with the paper:

First, the analysis is based on 2 hr 45 minutes and only 31 minutes for the two subspecies. Is this enough to characterize all clicks? Could there be other confounding factors, eg. Time of day influences on behavior, environmental impacts, etc. that would change the results if more data were used? The fact that you found that there were differences in the detection rate at the different frequencies among the subspecies in different habitats is evidence to me that there could be other factors that may impact the detection rates (e.g. time of day, etc.). These possible influences should be included in the discussion, and provide a stronger justification for the timing of the data that were used (e.g., time of day, environmental conditions impacting detection ability, etc.)

Secondly, counts were made using visual analysis of the full spectrogram, giving the counter the advantage of knowing whether a possible signal it is a part of a broader signal or not. However, the purpose or application of this paper was justified as being able to detect clicks from 2 different single-frequency recordings. Why weren't detections from the single-frequency methods proposed also tested? For example, it would be helpful to have an observer make counts from the full spectrograms, and compare to counts made from a subset of the data at the single frequencies. I expect that it is possible to get an over-estimate from the single-frequency counts since information is lost at the single frequency such as being able to tell if the signal is from a click or another source. This would only be applicable if dolphins are the only sound source for your area/time. But if there are any other sources (e.g., snapping shrimp – also very common and produces a broadband click, or fish, etc.) there would be no way of identifying the source of the signal. You must include some of these limitations and possible confounding factors in your discussion, after some thought about how applicable these single-frequency methods really would be.

Line 84 - "Data" is plural. Change data is/was to data are/were.

Review form: Reviewer 2

Is the manuscript scientifically sound in its present form?

Yes

Are the interpretations and conclusions justified by the results?

No

Is the language acceptable?

Yes

Do you have any ethical concerns with this paper?

No

Have you any concerns about statistical analyses in this paper?

No

Recommendation?

Accept with minor revision (please list in comments)

Comments to the Author(s)

This is a very nicely presented, well-organized study, but I'm a little worried about the acoustical analysis.

The authors don't say much about the detection and classification process, other than that "Only echolocation clicks were counted and identified." It would be helpful if they could to expand on that.

Were the detections done independently on the original and downsampled datasets, and then compared? That would be much better than, say, detecting the clicks in the 96 kHz data, and then determining which detections would have also been detected at other maximum frequency thresholds. To really simulate the effects of the different sample rates correctly, the data would have to be downsampled to 48 and 24 kHz, which affects temporal resolution even below the Nyquist/maximum viewable frequency. It is not enough to simply enforce different maximum frequency thresholds. Can you clarify which approach was used?

It would be helpful to know if the click detection was done in the time or frequency domain (I'm not familiar with Raven Pro's options), as that would shed some light on what exactly was going on in the acoustic analysis.

Assuming that the downsampling and detection part was done correctly, the authors then seem to be assuming an artificial case where the only signals detected are echolocation clicks, and only from one species. In that case, it might be reasonable to conclude that lower sampling rates are fine for this purpose.

However in reality, sampling rate probably also affects the ability to correctly distinguish echolocation clicks from other similar signals, and between species if more than one species is present. The more coarse the sampling rate, the harder it becomes to distinguish signals from each other, and in many cases the characteristic features used to make those distinctions are at higher frequencies. For instance, a bottlenose dolphin click and a snapping shrimp snap (a common source of false positives in coastal waters) could look similar at low frequencies, but one has more energy at higher frequencies than the other. Without that information, the distinction becomes more difficult.

To address this, false positive rates for the detector at different sample rates would be helpful, as would further info on how it was determined that these were dolphin clicks. Is that process scalable, or are these manual decisions?

I would ask that the authors address these issues/assumptions in the discussion, and as a caveat to their conclusions and abstract to avoid misleading readers as to the generalizability of these results, particularly if their analysis suggests that this is an issue.

Other comments:

- Figure 2c: The detection in the box looks like an 80kHz echosounder on a passing vessel, not dolphin. Hopefully that is just an error in the figure?

- Figure 2a & b: Why are the clicks in (a) so different in from (b)? Different location? Different species? Given the authors' conclusions, I expected to see examples of click trains in which some clicks were detected and others were missed at 48 vs 24kHz, but overall, the same click trains would be found. But here it looks more like one click type that would be detected, and one that would be missed. This implies that the different sample rates cannot be considered equivalent options.

- Table 1. Could the 96 kHz results be added to the table? The fact that they are not makes me wonder if these are really independent detection efforts. It is unclear to me what "Range of clicks detected" refers to. What are the units of these numbers?

Is it possible to say something about the received levels of the clicks detected at each sampling rate? Is there a pattern? (e.g. Maybe the clicks missed at lower sampling rates are lower amplitude?)

Decision letter (RSOS-201598.R0)

Dear Ms Romeu

The Editors assigned to your paper RSOS-201598 "Low frequency sampling rates are effective to detect bottlenose dolphins" have now received comments from reviewers and would like you to revise the paper in accordance with the reviewer comments and any comments from the Editors. Please note this decision does not guarantee eventual acceptance.

Please submit your revised manuscript and required files (see below) no later than 21 days from today's (ie 08-Feb-2021) date. Note: the ScholarOne system will 'lock' if submission of the revision is attempted 21 or more days after the deadline. If you do not think you will be able to meet this deadline please contact the editorial office immediately.

Best regards,
Lianne Parkhouse
Editorial Coordinator
Royal Society Open Science

on behalf of Professor Len Thomas (Associate Editor) and Pete Smith (Subject Editor)
openscience@royalsociety.org

Associate Editor Comments to Author (Professor Len Thomas):

Thank-you for your submission to RSOS. We have received two reviews from knowledgeable reviewers, both of whom raise substantial concerns. Reviewer 1 is concerned that there are too few data, and about your experimental protocol (that the counter had access to the full spectrogram). Reviewer 2 is concerned about the way you did the acoustic processing. I am therefore recommending possible acceptance pending major revisions. If you choose to send the manuscript back, acceptance will depend upon how you address these criticisms. Please also address the more minor comments made by the reviewers. In addition, please provide a small "Readme" file to accompany your data and code to help orient readers.

Reviewer comments to Author:

Reviewer: 1

Comments to the Author(s)

I have two major concerns with the paper:

First, the analysis is based on 2 hr 45 minutes and only 31 minutes for the two subspecies. Is this enough to characterize all clicks? Could there be other confounding factors, eg. Time of day influences on behavior, environmental impacts, etc. that would change the results if more data were used? The fact that you found that there were differences in the detection rate at the different frequencies among the subspecies in different habitats is evidence to me that there could be other factors that may impact the detection rates (e.g. time of day, etc.). These possible influences should be included in the discussion, and provide a stronger justification for the timing of the data that were used (e.g., time of day, environmental conditions impacting detection ability, etc.)

Secondly, counts were made using visual analysis of the full spectrogram, giving the counter the advantage of knowing whether a possible signal it is a part of a broader signal or not. However, the purpose or application of this paper was justified as being able to detect clicks from 2 different single-frequency recordings. Why weren't detections from the single-frequency methods proposed also tested? For example, it would be helpful to have an observer make counts from the full spectrograms, and compare to counts made from a subset of the data at the single frequencies. I expect that it is possible to get an over-estimate from the single-frequency counts since information is lost at the single frequency such as being able to tell if the signal is from a click or another source. This would only be applicable if dolphins are the only sound source for your area/time. But if there are any other sources (e.g., snapping shrimp – also very common and produces a broadband click, or fish, etc.) there would be no way of identifying the source of the signal. You must include some of these limitations and possible confounding factors in your discussion, after some thought about how applicable these single-frequency methods really would be.

Line 84 - "Data" is plural. Change data is/was to data are/were.

Reviewer: 2

Comments to the Author(s)

This is a very nicely presented, well-organized study, but I'm a little worried about the acoustical analysis.

The authors don't say much about the detection and classification process, other than that "Only echolocation clicks were counted and identified." It would be helpful if they could to expand on that.

Were the detections done independently on the original and downsampled datasets, and then compared? That would be much better than, say, detecting the clicks in the 96 kHz data, and then determining which detections would have also been detected at other maximum frequency thresholds. To really simulate the effects of the different sample rates correctly, the data would have to be downsampled to 48 and 24 kHz, which affects temporal resolution even below the Nyquist/maximum viewable frequency. It is not enough to simply enforce different maximum frequency thresholds. Can you clarify which approach was used?

It would be helpful to know if the click detection was done in the time or frequency domain (I'm not familiar with Raven Pro's options), as that would shed some light on what exactly was going on in the acoustic analysis.

Assuming that the downsampling and detection part was done correctly, the authors then seem to be assuming an artificial case where the only signals detected are echolocation clicks, and only from one species. In that case, it might be reasonable to conclude that lower sampling rates are fine for this purpose.

However in reality, sampling rate probably also affects the ability to correctly distinguish echolocation clicks from other similar signals, and between species if more than one species is present. The more coarse the sampling rate, the harder it becomes to distinguish signals from each other, and in many cases the characteristic features used to make those distinctions are at higher frequencies. For instance, a bottlenose dolphin click and a snapping shrimp snap (a common source of false positives in coastal waters) could look similar at low frequencies, but one has more energy at higher frequencies than the other. Without that information, the distinction becomes more difficult.

To address this, false positive rates for the detector at different sample rates would be helpful, as would further info on how it was determined that these were dolphin clicks. Is that process scalable, or are these manual decisions?

I would ask that the authors address these issues/assumptions in the discussion, and as a caveat to their conclusions and abstract to avoid misleading readers as to the generalizability of these results, particularly if their analysis suggests that this is an issue.

Other comments:

- Figure 2c: The detection in the box looks like an 80kHz echosounder on a passing vessel, not dolphin. Hopefully that is just an error in the figure?

- Figure 2a & b: Why are the clicks in (a) so different in from (b)? Different location? Different species? Given the authors' conclusions, I expected to see examples of click trains in which some clicks were detected and others were missed at 48 vs 24kHz, but overall, the same click trains would be found. But here it looks more like one click type that would be detected, and one that would be missed. This implies that the different sample rates cannot be considered equivalent options.

- Table 1. Could the 96 kHz results be added to the table? The fact that they are not makes me wonder if these are really independent detection efforts.

It is unclear to me what "Range of clicks detected" refers to. What are the units of these numbers?

Is it possible to say something about the received levels of the clicks detected at each sampling rate? Is there a pattern? (e.g. Maybe the clicks missed at lower sampling rates are lower amplitude?)

===PREPARING YOUR MANUSCRIPT===

===PREPARING YOUR REVISION IN SCHOLARONE===

<https://royalsociety.org/journals/authors/author-guidelines/#supplementary-material> to include a suitable title and informative caption. An example of appropriate titling and captioning may be found at https://figshare.com/articles/Table_S2_from_Is_there_a_trade-off_between_peak_performance_and_performance_breadth_across_temperatures_for_aerobic_sc_ope_in_teleost_fishes_/3843624.

Author's Response to Decision Letter for (RSOS-201598.R0)

See Appendix A.

RSOS-201598.R1 (Revision)

Review form: Reviewer 1

Is the manuscript scientifically sound in its present form?

Yes

Are the interpretations and conclusions justified by the results?

Yes

Is the language acceptable?

Yes

Do you have any ethical concerns with this paper?

No

Have you any concerns about statistical analyses in this paper?

No

Recommendation?

Accept as is

Comments to the Author(s)

This revised version has a greatly improved methods with much more detail that clearly described the analysis. I believe the issues described in the reviews from the first version have been adequately addressed and the paper now provides new, useful and interesting results that will be useful in the scientific field in setting up future passive recording studies. The previous issue of not having a direct comparison of the original sampling frequency to the unbiased downsampled frequencies has been resolved. Information on what sampling rates are necessary for different applications may allow for extended sampling deployments and improved sampling designs. The discussion now includes a stronger discussion of the limitations as well as the significance of the results of the research.

Review form: Reviewer 2

Is the manuscript scientifically sound in its present form?

Yes

Are the interpretations and conclusions justified by the results?

Yes

Is the language acceptable?

Yes

Do you have any ethical concerns with this paper?

No

Have you any concerns about statistical analyses in this paper?

No

Recommendation?

Accept as is

Comments to the Author(s)

Thank you for this very thorough and thoughtful revision. The paper is greatly improved, and I have no further comments.

Decision letter (RSOS-201598.R1)

Dear Ms Romeu,

It is a pleasure to accept your manuscript entitled "Low frequency sampling rates are effective to record bottlenose dolphins" in its current form for publication in Royal Society Open Science. The comments of the reviewer(s) who reviewed your manuscript are included at the foot of this letter.

on behalf of Professor Len Thomas (Associate Editor) and Pete Smith (Subject Editor)
openscience@royalsociety.org

Associate Editor Comments to Author (Professor Len Thomas):

Both reviewers thank the authors for their attention to earlier reviewer comments, and state that the manuscript is much improved. I concur and am happy to recommend acceptance.

Reviewer comments to Author:
Reviewer: 1
Comments to the Author(s)

This revised version has a greatly improved methods with much more detail that clearly described the analysis. I believe the issues described in the reviews from the first version have been adequately addressed and the paper now provides new, useful and interesting results that will be useful in the scientific field in setting up future passive recording studies. The previous issue of not having a direct comparison of the original sampling frequency to the unbiased

downsampled frequencies has been resolved. Information on what sampling rates are necessary for different applications may allow for extended sampling deployments and improved sampling designs. The discussion now includes a stronger discussion of the limitations as well as the significance of the results of the research.

Reviewer: 2

Comments to the Author(s)

Thank you for this very thorough and thoughtful revision. The paper is greatly improved, and I have no further comments.

Appendix A

Manuscript ID: RSOS-201598

Title: Low frequency sampling rates are effective to detect bottlenose dolphins

Authors: Bianca Romeu, Alexandre M. S. Machado, Fábio G. Daura-Jorge, Marta J. Cremer, Ana Kássia de Moraes Alves, Paulo C. Simões-Lopes

To the editorial board at Royal Society Open Science
Professor Len Thomas, Ph.D., Associate Editor

Dear Dr. Len Thomas,

We are grateful for the opportunity to revise and resubmit an improved version of our manuscript (RSOS-201598). We appreciate the time invested by the two anonymous referees in such attentive reviews, with thoughtful comments that helped us to improve and clarify our work. We particularly appreciated the Reviewer #2 statement that “this is a very nicely presented, well-organized study”.

We reviewed the manuscript thoroughly to address all the comments carefully. We paid special attention to two major concerns: our sample size and the acoustic processing. Based on the insightful comments from both reviewers, we decided to take a step back and re-analyse the data using a different framework. Therefore, in this reviewed version, we add a new set of results using a more suitable approach to our work, downsampling each recording, as suggested by reviewers. We highlight that, despite our new approach and other methodological changes, the final results and conclusions of our manuscript remained the same. We address all these changes in our point-by-point response letter, and we adjusted the main text to clarify how we now carried out the acoustic processing.

Here, we resubmit a new version of our manuscript for your consideration for publication in the journal Royal Society Open Science. Please, find attached to this resubmission the responses to each of the reviewers' comments. For your convenience, we uploaded both a clean and a tracked changes version of the manuscript.

On behalf of all co-authors, I would like to thank you and the reviewers for the attentive and constructive review. We are looking forward to hearing from you.

Yours sincerely,

Bianca Romeu & co-authors

Laboratório de Mamíferos Aquáticos
Departamento de Ecologia e Zoologia
Universidade Federal de Santa Catarina
bianca.romeu@posgrad.ufsc.br

Reviewer #1 comments to Authors:

Comment #1 by Reviewer #1: I have two major concerns with the paper:

First, the analysis is based on 2 hr 45 minutes and only 31 minutes for the two subspecies. Is this enough to characterize all clicks? Could there be other confounding factors, eg. Time of day influences on behavior, environmental impacts, etc. that would change the results if more data were used? The fact that you found that there were differences in the detection rate at the different frequencies among the subspecies in different habitats is evidence to me that there could be other factors that may impact the detection rates (e.g. time of day, etc.). These possible influences should be included in the discussion, and provide a stronger justification for the timing of the data that were used (e.g., time of day, environmental conditions impacting detection ability, etc.)

Response: We are grateful for your attentive review. We addressed each of your comments below and followed all specific suggestions to improve the presentation of our work. Note that line numbers refer to the revised version of the manuscript with inline tracked changes.

We understand the reviewer's concern regarding the time of recordings and the confounding factors, but we have both biological and statistical reasons to support the robustness of our findings.

The recordings of *Tursiops truncatus truncatus* were made opportunistically as now we commented in the main text (please, see line 111). However, despite the small sample size, we decided to use these records to make an initial comparison of the probability of record echolocation clicks at different frequencies between different subspecies and environments. We agree that with this small sample size, we have some limitations to reach strong conclusions, but we still found this comparison useful for the context of our study. Therefore, in the discussion section, we recognize the limitations related to the small sample size but justified why we kept this comparison considering the aims of our study and highlighted more recording would be need for better conclusions (please, see lines 391-405).

*Lines 391-405: "We decided to analyze two subspecies in two different environments, even with few records of *T. t. truncatus*, because the acoustic parameters of echolocation clicks can vary between different *Tursiops* species [51] and environmental characteristics can affect sound production and propagation [5,52,53]. However, we were unable to distinguish whether the reduction in the occurrence of echolocation clicks from *T. t. truncatus* below 24 kHz is due to environmental variations, differences in sound emissions between subspecies, or both, because each subspecies inhabits a different environment and can adapt their sound emissions according to environmental characteristics [54]. Furthermore, the random effects of our models show that other factors than the subspecies and the environment can also explain such variations in the frequency range of echolocation clicks. Many of these factors cannot be controlled when we record free-living animals, such as the distance and position of a moving dolphin to the hydrophone [55]. However, more samples from *T. t. truncatus* can help us to understand this pattern, as well as consider other variables such as behavior, group size and composition, environmental conditions, etc."*

In addition, although the time of recordings is small, we have 15 different groups recorded for the coastal and 3 different groups recorded for the offshore subspecies. Such different recordings comprise different environmental conditions, behavioural differences, depth, dolphins' distance and position to the hydrophone—to name a few. Some of these factors are still challenging to assess in the field, such as the dolphins' position and distance to the hydrophone (see Au and Benoit-Bird, 2003). However, note that we already acknowledge some of these confounding factors that we accounted for in the analysis—for example, see lines 242-245. Also, note that we quantified the influence of such factors. More specifically, we used Generalized Linear Mixed Models, in which we defined pairs of 1-minute fraction samples nested to the respective recording as the random intercepts. This structure of random effects allowed us to account for the confounding factors during any given recording (lines 238-231).

Note that we already reported measures of goodness of fit for the fixed effects only (Marginal $R^2 = 56.70\%$) and for both fixed and random effects (Conditional $R^2 = 99.94\%$) (lines 296-298). The random effects account for ca. 43% of the variation in acoustic detections but determining which of the confounding factors underpins such variation in acoustic detection is still unfeasible for most in situ studies.

We now added and adjusted the text to acknowledge the effect of other confounding factors and highlight how we accounted for that in our methods. Please, see lines 391-405 copied above.

Comment #2 by Reviewer #1: Secondly, counts were made using visual analysis of the full spectrogram, giving the counter the advantage of knowing whether a possible signal it is a part of a broader signal or not. However, the purpose or application of this paper was justified as being able to detect clicks from 2 different single-frequency recordings. Why weren't detections from the single-frequency methods proposed also tested? For example, it would be helpful to have an observer make counts from the full spectrograms, and compare to counts made from a subset of the data at the single frequencies. I expect that it is possible to get an over-estimate from the single-frequency counts since information is lost at the single frequency such as being able to tell if the signal is from a click or another source. This would only be applicable if dolphins are the only sound source for your area/time. But if there are any other sources (e.g., snapping shrimp – also very common and produces a broadband click, or fish, etc.) there would be no way of identifying the source of the signal. You must include some of these limitations and possible confounding factors in your discussion, after some thought about how applicable these single-frequency methods really would be.

Response: The reviewer is correct. Thank you for these comments. The analysis of the full frequency range tends to over-estimate the echolocation clicks record in lower frequencies when recordings are made using lower-frequency sampling rates. Therefore, we changed our approach concerning the visual analysis of the full frequency range, and we added a new approach to confirm the echolocation clicks detection in the lower frequencies. We detailed our new approach in *Methods* session (see lines 131-138, 173-218, and 247-259), and we adjusted the text according in *Results* and *Discussion* sessions (see lines 268-270, 323-338, 358-379).

We also added some considerations and caveats in the discussion about using lower frequencies to investigate echolocation clicks. Please, see lines 434-440.

Lines 434-440: *"Here, we showed that lower frequencies record bottlenose dolphins' echolocation clicks efficiently for a set of ecological studies. In other words, studies that investigate the occurrence or distribution of individuals in a population, when presence/absence data are sufficient, can use recordings at low frequencies. However, it is necessary to highlight, when the goal is to analyze acoustic parameters of clicks or identify different dolphins' species, it is recommended to record, as much as possible, all clicks frequency ranges [51,65]. Also, it should consider the soundscape of the study area, thinking about possible solutions to the acoustic data analysis."*

Comment #3 by Reviewer #1: Line 84 - "Data" is plural. Change data is/was to data are/were.

Response: Thank you for this comment. We changed this sentence that now reads:

Line 105: *"The data were collected from December 4th to 12th (...)"*.

Reviewer #2 comments to Authors:

Comment #1 by Reviewer #2: This is a very nicely presented, well-organized study, but I'm a little worried about the acoustical analysis.

Response: Thank you for considering these qualities in our work. We appreciate your time invested in the review, helping us to improve our work. We considered all your concerns about our acoustical analysis and we carefully reanalysed the data taking into account your comments and suggestions. Please, see our responses to specific comments below. Note that line numbers refer to the revised version of the manuscript with inline tracked changes.

Comment #2 by Reviewer #2: The authors don't say much about the detection and classification process, other than that "Only echolocation clicks were counted and identified." It would be helpful if they could to expand on that.

Response: Excuse the lack of clarity. We reorganized and added information about the initial analysis approach that we use. The text now reads:

Lines 148-160: *"The spectrograms were visually inspected to identify echolocation clicks. These clicks were defined and identified as those belonging to click trains with interclick intervals longer than 10 ms [9,35,36]. The total number of echolocation clicks recorded up to 96 kHz was counted manually. Each echolocation click was visually inspected to verify its occurrence below and/or above frequency thresholds of 24 kHz and 48 kHz given the full frequency range of the recordings. Then clicks were counted in each frequency threshold to estimate the proportion of clicks that appear in each threshold. The total number of echolocation clicks recorded up to 96 kHz was paired with the total number of echolocation clicks that occurred in 24 kHz and/or 48 kHz thresholds (Fig. 2). This is because the same click, when visualized below 24 kHz, was counted as "up to 24 kHz" and "up to 48 kHz" (Fig. 2, the solid line brown rectangles), while the clicks with frequencies above 24 kHz, but below 48 kHz, were only counted as "up to 48 kHz" (Fig. 2a, the dashed line brown rectangle)."*

Comment #3 by Reviewer #2: Were the detections done independently on the original and downsampled datasets, and then compared? That would be much better than, say, detecting the clicks in the 96 kHz data, and then determining which detections would have also been detected at other maximum frequency thresholds. To really simulate the effects of the different sample rates correctly, the data would have to be downsampled to 48 and 24 kHz, which affects temporal resolution even below the Nyquist/maximum viewable frequency. It is not enough to simply enforce different maximum frequency thresholds. Can you clarify which approach was used?

Response: Thank you for this comment that makes us rethink how we processed our data and counted the clicks. Initially, we indeed have made the counts looking for echolocation clicks just at the original frequency range of 96 kHz, and then we observed the echolocation clicks below 24 and 48 kHz. We agree, however, that this was not the better approach to confirm the record of these signals at different frequencies. Thus, based on your comments, we decided to take a step back and re-analyse the data in downsampled recordings too. We described this new approach and other changes in *Method* session (lines 173-218) and adjusted the text throughout the manuscript to align it with our new approach. Main changes were in the following lines:

Lines 131-138: *"Two different approaches were used to analyze if echolocation clicks recorded in a frequency range up to 96 kHz can be recorded in frequency ranges below 24 and 48 kHz. First, the recordings were assessed visually through spectrograms to quantify the proportion of echolocation clicks occurring in each frequency range of interest. That is, each click observed in the spectrograms was verified if it occurred below and above frequency thresholds of 24 and 48 kHz. Second, the recordings were downsampled at 48 and 96 kHz (Nyquist frequency of 24 and 48 kHz, respectively) and*

processed by an automatic signal detector, to test if echolocation clicks remain detectable when recordings are made at a lower sampling rate.”

Lines 173-218: “A sampled high-frequency signal can change when it is sampled in different, lower sampling rates [30]. Because of this, in our second approach, the recordings were downsampled at 48 and 96 kHz to analyze if echolocation clicks remain detectable at 24 and 48 kHz. The downsample and the following echolocation clicks automatic detection were made using R 3.6.0 [37]. First, a fourth-order Butterworth 15 kHz high-pass filter and an anti-aliasing Finite Impulse Response (FIR) low-pass filter to the 24 or 48 kHz were applied in the original recordings, for each corresponding downsample frequency. Then, the downsample was made using the “downsample” function of the “tuneR” R package [38]. Next, we used the “auto_detec” function of the “warbleR” R package [39] to detect the echolocation clicks automatically in recordings at 96, 48, and 24 kHz—in other words, the original and downsampled frequencies. The parameters to configure the “auto_detec” were defined through the “optimize_auto_detec” function from the “warbleR” R package [39].”

We also added considerations about the complementary approach in the *Discussion* session (lines 358-379):

Lines 358-379: “Our estimate of the proportion of echolocation clicks recorded with frequency ranges below 24 and 48 kHz shows that these clicks can be recorded at lower frequencies. However, this proportion cannot be extrapolated to the echolocation clicks recorded at lower sampling rates. When a high-frequency signal is sampled with a low sampling rate, the sampled signal can be different from the real signal [30]. Then, we downsampled the recordings to simulate recordings made with lower sampling rates and confirm that echolocation clicks are still preserved in these frequencies. Therefore, unfortunately, we cannot present a definitive evaluation of the performance of lower sampling rates in recording echolocation clicks because of the effect of three main reasons discussed below: frequencies evaluated, soundscape, and detector performance.

Downsampling the original files at 48 and 96 kHz sampling rates increase the soundscape interference in the signal detection. Snapping shrimp snaps, for example, are commonly present in the soundscape, and their sound is similar to dolphin clicks [46–49]. In our study area, mainly in the lagoon system, snapping shrimp snaps are part of the soundscape. Then, when we used the automatic detector in the downsampled recordings, especially at 24 kHz, the detection performance was limited. It is not difficult to differentiate the echolocation clicks from the snapping shrimp snaps when visually inspecting spectrograms. The former has patterns in inter-clicks intervals [46]. However, the soundscape composition can limit the use of an automatic detector [50]. To avoid counting shrimp snaps, we could exclude as much as possible the false positive detections using some addition steps in our processing protocol. However, there are not a definitive solution to this problem [50], and the use of automatic detector at lower sampling rates has some limitations.”

Comment #4 by Reviewer #2: It would be helpful to know if the click detection was done in the time or frequency domain (I'm not familiar with Raven Pro's options), as that would shed some light on what exactly was going on in the acoustic analysis.

Response: We made the visual analysis using spectrograms in Raven, which uses Fourier transformation to create them. Therefore, we analysed the recordings in the frequency domain. This information was added in lines 146-147.

Lines 146-147: “Raven uses Fourier transform to create spectrograms as a frequency domain representation of the signal.”

Comment #5 by Reviewer #2: Assuming that the downsampling and detection part was done correctly, the authors then seem to be assuming an artificial case where the only signals detected are echolocation clicks, and only from one species. In that case, it might be reasonable to conclude that lower sampling rates are fine for this purpose.

Response: Thank you for the opportunity to clarify our conclusions. First, as aforementioned, we now added a new approach downsampling the recordings at 48 and 96 kHz (maximum frequency of 24 and 48 kHz, respectively) and processed it by an automatic signal detector.

We are aware that the soundscape is composed of many other signals, from biological and anthropogenic sources, including in our main study area, the lagoon system. This might be indeed a problem to identify and detect precisely echolocation clicks. But this does not mean that echolocation clicks are not present and detectable, and that was one of the main purposes of our work: to show that it is possible to record echolocation clicks even in lower frequencies. For example, in low frequencies like 24 kHz, we can identify echolocation clicks by their inter-click intervals. However, we agree that in some soundscape precisely identifying dolphin's clicks is a challenge, especially in low frequencies and using automatic detectors. Regarding this, we added a few considerations in lines 368-379.

Lines 368-379: "Downsampling the original files at 48 and 96 kHz sampling rates increase the soundscape interference in the signal detection. Snapping shrimp snaps, for example, are commonly present in the soundscape, and their sound is similar to dolphin clicks [46–49]. In our study area, mainly in the lagoon system, snapping shrimp snaps are part of the soundscape. Then, when we used the automatic detector in the downsampled recordings, especially at 24 kHz, the detection performance was limited. It is not difficult to differentiate the echolocation clicks from the snapping shrimp snaps when visually inspecting spectrograms. The former has patterns in inter-clicks intervals [46]. However, the soundscape composition can limit the use of an automatic detector [50]. To avoid counting shrimp snaps, we could exclude as much as possible the false positive detections using some addition steps in our processing protocol. However, there are not a definitive solution to this problem [50], and the use of automatic detector at lower sampling rates has some limitations."

In addition, based on this and maybe other referees' comments, we believe the use of the work "detect" was misinforming the main objectives of our study, that was test the use of different frequencies to record of clicks, and based on these recordings, infer on the presence of dolphin in the study areas. We therefore replaced the word 'detect' by 'record' in many parts of the text, including the title.

Regarding the occurrence of more than one dolphin species at the same area, we highlight that this is an issue beyond the scope of our study. We have no intention to identify dolphin species, but record the presence of dolphins by recording echolocation clicks using low frequencies sampling rates. In our study, the coastal subspecies *T. t. gephyreus* were recorded in an area where with no occurrence of other species. However, if a study case is using PAM and intend to investigate presence/absence data of a specific species based on acoustic recordings, the presence of other species should be taking into account, and the use of all click's frequency ranges would be recommended. We clarified the intention of your study in lines 92-96 and we commented the possibility of recording others species in lines 435-440.

Lines 92-96: "Here, we evaluated whether the echolocation clicks produced by the bottlenose dolphins can be effectively recorded in frequencies up to 24 and 48 kHz. We aim to promote the use of PAM in ecological research on bottlenose dolphins, because it is a useful tool that can inform conservation and management plans, especially in countries where science budgets are limited."

Lines 435-440: "... studies that investigate the occurrence or distribution of individuals in a population, when presence/absence data are sufficient, can use recordings at low frequencies. However, it is necessary to highlight, when the goal is to analyze acoustic

parameters of clicks or identify different dolphins' species, it is recommended to record, as much as possible, all clicks frequency ranges [51,65]. Also, it should consider the soundscape of the study area, thinking about possible solutions to the acoustic data analysis."

Comment #6 by Reviewer #2: However, in reality, sampling rate probably also affects the ability to correctly distinguish echolocation clicks from other similar signals, and between species if more than one species is present. The more coarse the sampling rate, the harder it becomes to distinguish signals from each other, and in many cases the characteristic features used to make those distinctions are at higher frequencies. For instance, a bottlenose dolphin click and a snapping shrimp snap (a common source of false positives in coastal waters) could look similar at low frequencies, but one has more energy at higher frequencies than the other. Without that information, the distinction becomes more difficult.

Response: We agree with your comment. We are aware of this problem, which was more evident when we looked for an automatic detector. However, it was not our intention with this work to solve the problem of the distinction between dolphins' clicks and snapping shrimp snap. As aforementioned, the goal of our work is to show that it is possible to record echolocation clicks in lower frequencies. These clicks cannot be recognized individually if it is isolated from others, but echolocation clicks occur in click trains. And it is not difficult to recognize an echolocation click train, which is sufficient to identify the presence of the dolphin during the recording. In addition, it seems that the word "detection", used to express the presence of the signal, confused regarding the goal of our work. We replaced this word in the title and the text by the word 'record'. We also added, in the discussion, the limitations about echolocation click detection and differentiation between dolphin species when low frequencies are used (lines 368-419 and 417-419).

Lines 368-379: Please, see response to comment #5.

Lines 417-419: *"However, it is necessary to highlight, when the goal is to analyze acoustic parameters of clicks or identify different dolphins' species, it is recommended to record, as much as possible, all clicks frequency ranges [51,65]."*

Comment #7 by Reviewer #2: To address this, false positive rates for the detector at different sample rates would be helpful, as would further info on how it was determined that these were dolphin clicks. Is that process scalable, or are these manual decisions?

Response: Excuse for the lack of clarity. Initially, we did not use an automatic detector, but visually analysed spectrograms looking for echolocation clicks (line 148). After your comments, we used an automatic detector to confirm that echolocation clicks are recorded in low frequencies. We could not find an automatic detector with good precision to detect echolocation clicks, partially due to the snapping shrimps snap in our recordings. We recognized and explained this issue in lines 370-379.

Lines 370-379: *"In our study area, mainly in the lagoon system, snapping shrimp snaps are part of the soundscape. Then, when we used the automatic detector in the downsampled recordings, especially at 24 kHz, the detection performance was limited. It is not difficult to differentiate the echolocation clicks from the snapping shrimp snaps when visually inspecting spectrograms. The former has patterns in inter-clicks intervals [46]. However, the soundscape composition can limit the use of an automatic detector [50]. To avoid counting shrimp snaps, we could exclude as much as possible the false positive detections using some addition steps in our processing protocol. However, there are not a definitive solution to this problem [50], and the use of automatic detector at lower sampling rates has some limitations."*

Regarding false positives, we proceed with some tests and concluded that, for the automatic detector we used, the false positive rates are not the same to all recordings (lines 196-198). We did not measure the detector performance in each sample rate. We used some strategies to reduce the possibility of false positives be considered in our analysis (see lines 199-210). Detections at the frequency of 48 kHz are better than at the frequency of 24 kHz, as expected.

Lines 196-198: "Even though we have selected the optimal parameters to configure the signal detector based on a priori manual detections, the sensibility and specificity of the detector were not the same to all recordings."

Lines 199-210: "Furthermore, additional filters were applied to reduce the number of false-positive detections. First, all detections were filtered based on the interval of signals detected, similar to interclick intervals [see 40], excluding detections with an interval longer than 0.2 s, since echolocation clicks occur in click trains, not isolated. Second, the detection localization (in time) in the original recording was compared with detection localization in downsampled recordings, assuming the detector performance was better in recordings at 96 kHz. Since the echolocation clicks in the downsampled recordings tend to be longer than in the original record, a buffer was created around each detection of the original recordings by expanding the start and end time of detections by the length of each signal (i.e. click duration) to guarantee the match between the same detections at each frequency. Next, recordings were divided into bins of one second and matching bins from the division of the 1-minute samples from the first approach."

Comment #8 by Reviewer #2: I would ask that the authors address these issues/assumptions in the discussion, and as a caveat to their conclusions and abstract to avoid misleading readers as to the generalizability of these results, particularly if their analysis suggests that this is an issue.

Response: Thank you for your review. We carefully think about all your comments and we made many changes to improve and clarify our results and conclusions. Besides changing the methods, results, and discussion, as reported above in each response to respective comments, we change the title and de abstract (lines 23-24 and 27-31) to avoid misinterpretation regarding our goal with and results of this work. We believe that after account for all these valuable comments, our manuscript improved significantly.

Other comments:

Comment #9 by Reviewer #2: Figure 2c: The detection in the box looks like an 80kHz echosounder on a passing vessel, not dolphin. Hopefully that is just an error in the figure?

Response: Thank you for perceiving this mistake. As the recording represented in the figure was made into the lagoon system, we did not think that it could be a kind of sounder. We changed the figure (line 166) and excluded the recordings with this signal from the analysis.

Comment #10 by Reviewer #2: - Figure 2a & b: Why are the clicks in (a) so different in from (b)? Different location? Different species? Given the authors' conclusions, I expected to see examples of click trains in which some clicks were detected and others were missed at 48 vs 24kHz, but overall, the same click trains would be found. But here it looks more like one click type that would be detected, and one that would be missed. This implies that the different sample rates cannot be considered equivalent options.

Response: Thanks for this question. The reviewer is right in your conclusion regarding click detection in click trains. We did not think about this aspect when selected the spectrograms to illustrate the manuscript. Now, we change the figure (line 166).

Comment #11 by Reviewer #2: - Table 1. Could the 96 kHz results be added to the table? The fact that they are not makes me wonder if these are really independent detection efforts. It is unclear to me what "Range of clicks detected" refers to. What are the units of these numbers?

Response: We changed the table in lines 272-278 to make it clearer.

Comment #12 by Reviewer #2: Is it possible to say something about the received levels of the clicks detected at each sampling rate? Is there a pattern? (e.g. Maybe the clicks missed at lower sampling rates are lower amplitude?)

Response: Since we recorded free-living animals, it is not possible to know the dolphins' distance and position to the hydrophone. Thus, we cannot say what was the received levels of the clicks recorded, which was much variable. Probably some missed clicks at lower frequencies are due to lower amplitude.